# MAgNet: Mesh Agnostic Neural PDE Solver

**Oussama Boussif**
Mila - Québec AI Institute
DIRO, Université de Montréal
oussama.boussif@mila.quebec

**Dan Assouline**
Mila - Québec AI Institute
DIRO, Université de Montréal
dan.assouline@mila.quebec

**Loubna Benabbou**
Université du Québec à Rimouski
Loubna_Benabbou@uqar.ca

**Yoshua Bengio**[*]
Mila - Québec AI Institute
DIRO, Université de Montréal
yoshua.bengio@mila.quebec

## Abstract

The computational complexity of classical numerical methods for solving Partial Differential Equations (PDE) scales significantly as the resolution increases. As an important example, climate predictions require fine spatio-temporal resolutions to resolve all turbulent scales in the fluid simulations. This makes the task of accurately resolving these scales computationally out of reach even with modern supercomputers. As a result, current numerical modelers solve PDEs on grids that are too coarse (3km to 200km on each side), which hinders the accuracy and usefulness of the predictions. In this paper, we leverage the recent advances in Implicit Neural Representations (INR) to design a novel architecture that predicts the spatially continuous solution of a PDE given a spatial position query. By augmenting coordinate-based architectures with Graph Neural Networks (GNN), we enable zero-shot generalization to new non-uniform meshes and long-term predictions up to 250 frames ahead that are physically consistent. Our Mesh Agnostic Neural PDE Solver (MAgNet) is able to make accurate predictions across a variety of PDE simulation datasets and compares favorably with existing baselines. Moreover, MAgNet generalizes well to different meshes and resolutions up to four times those trained on[2].

## 1   Introduction

Partial Differential Equations (PDEs) describe the continuous evolution of multiple variables, e.g. over time and/or space. They arise everywhere in physics, from quantum mechanics to heat transfer and have several engineering applications in fluid and solid mechanics. However, most PDEs can't be solved analytically, so it is necessary to resort to numerical methods. Since the introduction of computers, many numerical approximations were implemented, and new fields emerged such as Computational Fluid Mechanics (CFD) (Richardson and Lynch, 2007). The most famous numerical approximation scheme is the Finite Element Method (FEM) (Courant, 1943; Hrennikoff, 1941). In the FEM, the PDE is discretized along with its domain, and the problem is transformed into solving a set of matrix equations. However, the computational complexity scales significantly with the resolution. For climate predictions, this number can be quite significant if the desired error is to be reached, which renders its use impractical.

---

[*]CIFAR Senior Fellow

[2]Code and dataset can be found on: `https://github.com/jaggbow/magnet`

36th Conference on Neural Information Processing Systems (NeurIPS 2022).

In this paper, we propose to learn the *continuous* solutions for spatio-temporal PDEs. Previous methods focused on either generating fixed resolution predictions or generating arbitrary resolution solutions on a fixed grid (Li et al., 2021; Wang et al., 2020). PDE models based on Multi-Layer Perceptrons (MLPs) can generate solutions at any point of the domain (Dissanayake and Phan-Thien, 1994; Lagaris et al., 1998; Raissi et al., 2017a). However, without imposing a physics-motivated loss that constrains the predictions to follow the smoothness bias resulting from the PDE, MLPs become less competitive than CNN-based approaches especially when the PDE solutions have high-frequency information (Rahaman et al., 2018).

We leverage the recent advances in Implicit Neural Representations ((Tancik et al., 2020), (Chen et al., 2020), (Jiang et al., 2020)) and propose a general purpose model that can not only learn solutions to a PDE with a resolution it was trained on, but it can also perform zero-shot super-resolution on irregular meshes. The added advantage is that we propose a general framework where we can make predictions given any spatial position query for both grid-based architectures like CNNs and graph-based ones able to handle sensors and predictions at arbitrary spatial positions.

**Contributions** Our main contributions are in the context of machine learning for approximately but efficiently solving PDEs and can be summarized as follows:

- We propose a framework that enables grid-based and graph-based architectures to generate continuous-space PDE solutions given a spatial query at any position.
- We show experimentally that this approach can generalize to resolutions up to four times those seen during training in zero-shot super-resolution tasks.

## 2 Related Works

Current solvers can require a lot of computations to generate solutions on a fine spatio-temporal grid. For example, climate predictions typically use General Circulation Models (GCM) to make forecasts that span several decades over the whole planet (Phillips, 1956). These GCMs use PDEs to model the climate in the atmosphere-ocean-land system and to solve these PDEs, classical numerical solvers are used. However, the quality of predictions is bottlenecked by the grid resolution that is in turn constrained by the available amount of computing power. Deep learning has recently emerged as an alternative to these classical solvers in hopes of generating data-driven predictions faster and making approximations that do not just rely on lower resolution grids but also on the statistical regularities that underlie the family of PDEs being considered. Using deep learning also makes it possible to combine the information in actual sensor data with the physical assumptions embedded in the classical PDEs. All of this would enable practitioners to increase the actual resolution further for the same computational budget, which in turn improves the quality of the predictions.

**Machine Learning for PDE solving:** Dissanayake and Phan-Thien (1994) published one of the first papers on PDE solving using neural networks. They parameterized the solutions to the Poisson and heat transfer equations using an MLP and studied the evolution of the error with the mesh size. Lagaris et al. (1998) used MLPs for solving PDEs and ordinary differential equations. They wrote the solution as a sum of two components where the first term satisfies boundary conditions and is not learnable, and the second is parameterized with an MLP and trained to satisfy the equations. In Raissi et al. (2017a) the authors also parameterized the solution to a PDE using an MLP that takes coordinates as input. With the help of automatic differentiation, they calculate the PDE residual and use its MSE loss along with an MSE loss on the boundary conditions. In follow-up work, Raissi et al. (2017b) also learn the parameters of the PDE (e.g. Reynolds number for Navier-Stokes equations).

The recently introduced Neural Operators framework (Kovachki et al., 2021; Li et al., 2020b,a) attempts to learn operators between spaces of functions. Li et al. (2021) use "Fourier Layers" to learn the solution to a PDE by framing the problem as learning an operator from the space of initial conditions to the space of the PDE solutions. Their model can learn the solution to PDEs that lie on a uniform grid while maintaining their performance in the zero-shot super-resolution setting. In the same spirit, Jiang et al. (2020) developed a model based on Implicit Neural Representations called "MeshFreeFlowNet" where they upsample existing PDE solutions to a higher resolution. They use 3D low-resolution space-time tensors as inputs to a 3DUnet in order to generate a feature map. Next, some points are sampled uniformly from the corresponding high-resolution tensors and fed to an MLP called ImNet (Chen and Zhang, 2018). They train their model using a PDE residual loss and

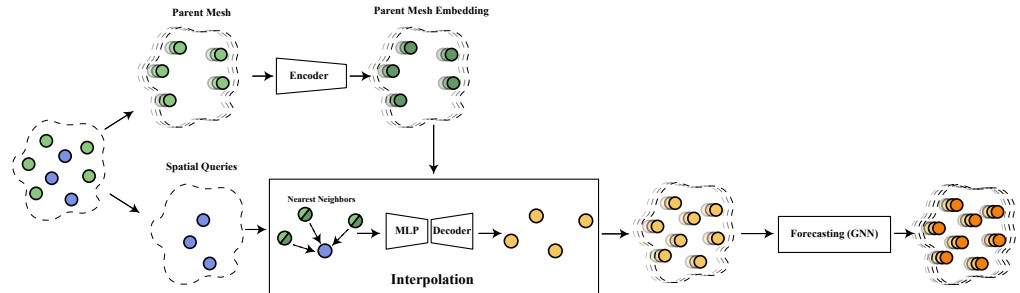

Figure 1: We illustrate the "Encode-Interpolate-Forecast" framework of MAgNet. The **parent mesh** is fed to the encoder to generate the **parent mesh embedding**. Next, we estimate the values at the **spatial queries** using the interpolation module that uses features from both the **parent mesh** points and the **parent mesh embedding** points closest to these queries. Finally, the **parent mesh** observations and interpolated values at **spatial queries** are gathered as nodes forming a **new graph** using nearest neighbors and the PDE solution is forecast for all nodes (therefore all spatial locations) into the **future** using the forecasting module.

are able to predict the flow field at any spatio-temporal coordinate. Their approach is closest to the one we propose here. The main difference is that we perform super-resolution on the spatial queries and forecast the solution to a PDE instead of only doing super-resolution on the existing sequence.

Brandstetter et al. (2022) use the message-passing paradigm ((Gilmer et al., 2017), (Watters et al., 2017), (Sanchez-Gonzalez et al., 2020)) to solve 1D PDEs. They are able to beat state-of-the-art Fourier Neural Operators (Li et al., 2021) and classical WENO5 solvers while introducing the "pushforward trick" that allows them to generate better long-term rollouts. Moreover, they present an added advantage over existing methods since they can learn PDE solutions at any mesh. However, they are not able to generalize to different resolutions, which is a crucial capability of our method.

Most machine learning approaches require data from a simulator in order to learn the required PDE solutions and that can be expensive depending on the PDE and the resolution. Wandel et al. (2020) alleviate that requirement by using a PDE loss.

**Machine Learning for Turbulence Modeling:** Recent years have known a surge in machine learning-based models for modeling turbulence. Since it is expensive to resolve all relevant scales, some methods were developed that only solve large scales explicitly and separately model sub-grid scales (SGS). Recently, Novati et al. (2021) used multi-agent reinforcement learning to learn the dissipation coefficient of the Smagorinsky SGS model (Smagorinsky, 1963) using as reward the recovery of the statistical properties of Direct Numerical Simulations (DNS). Rasp et al. (2018) used MLPs to represent sub-grid processes in clouds and replace previous parametrization models in a global general circulation model. In the same fashion, Park and Choi (2021) used MLPs to learn DNS sub-grid scale (SGS) stresses using as input filtered flow variables in a turbulent channel flow. Brenowitz and Bretherton (2018) use MLPs to predicts the apparent sources of heat and moisture using coarse-grained data and use a multi-step loss to optimize their model.Wang et al. (2020) used one-layer CNNs to learn the spatial filter in LES methods and the temporal filter in RANS as well as the turbulent terms. A UNet (Ronneberger et al., 2015) is then used as a decoder to get the flow velocity. de Bezenac et al. (2017) predict future frames by deforming the input sequence according to the advection-diffusion equation and apply it to Sea-Surface Temperature forecasting.

Stachenfeld et al. (2021) use the "encode-process-decode" (Sanchez-Gonzalez et al., 2018, 2020) paradigm along with dilated convolutional networks to capture turbulent dynamics seen in high-resolution solutions only by training on low spatial and temporal resolutions. Their approach beats existing neural PDE solvers in addition to the state-of-the-art `Athena++` engine (Stone et al., 2020). We take inspiration of this approach but replace the process module by an interpolation module, to allow the model to capture spatial correlations between known points and new query points.

## 3 Methodology

We present the developed framework that leverages recent advances in Implicit Neural Representations (INR) (Jiang et al., 2020; Sitzmann et al., 2020; Chen et al., 2020; Tancik et al., 2020) and draws

inspiration from mesh-free methods for PDE solving. We first start by giving a mathematical definition of a PDE. Next, we showcase the proposed "MAgNet" and derive two variants: A grid-based architecture and a graph-based one.

## 3.1 Preliminaries

We define PDE as follows, using $D^k$ to denote $k$-th order derivatives:

**Definition 3.1.** Evans (2010) *Let $U$ denote an open subset of $\mathbb{R}^n$ and $k \geq 1$ an integer. An expression of the form*:

$$\mathcal{L}(D^k \mathbf{u}(x), D^{k-1}\mathbf{u}(x), \ldots, \mathbf{u}(x), x) = \mathbf{0} \quad \forall x \in U \tag{1}$$

*is called a $k$-th order system of PDEs, where $\mathcal{L} : \mathbb{R}^{mn^k} \times \mathbb{R}^{mn^{k-1}} \times \cdots \times \mathbb{R}^{mn} \times \mathbb{R}^m \times U \to \mathbb{R}^m$ is given and $\mathbf{u} : U \to \mathbb{R}^m, \mathbf{u} = (u^1, \ldots, u^m)$ is the unknown function to be characterized.*

In this paper, we are interested in spatio-temporal PDEs. In this class of PDEs, the domain is $U = [0, +\infty] \times \mathcal{S}$ (time $\times$ space) where $\mathcal{S} \subset \mathbb{R}^n, \quad n \geq 1$ and, with $D^k$ indicating differentiation w.r.t $x$, any such PDE can be formulated as:

$$\begin{cases} \frac{\partial \mathbf{u}}{\partial t} = \mathcal{L}(D^k \mathbf{u}(x), \ldots, \mathbf{u}(x), x, t) & \forall t \geq 0, \forall x \in \mathcal{S}. \\ u(0, x) = g(x) & \forall x \in \mathcal{S} \\ \mathcal{B}u = 0 & \forall t \geq 0, \forall x \in \partial\mathcal{S} \end{cases} \tag{2}$$

Where $\partial\mathcal{S}$ is the boundary of $\mathcal{S}$, $\mathcal{B}$ is a non-linear operator enforcing boundary conditions on $u$ and $g : \mathcal{S} \to \mathbb{R}^m$ represents the initial condition constraints for the solution $u$.

Numerical PDE simulations have enjoyed a great body of innovations especially where their use is paramount in industrial applications and research. Mesh-based methods like the FEM numerically compute the PDE solution on a predefined mesh. However, when there are regions in the PDE domain that present large discontinuities, the mesh needs to be modified and provided with many more points around that region in order to obtain acceptable approximations. Mesh-based methods typically solve this problem by re-meshing in what is called Adaptive Mesh Refinement (Berger and Oliger, 1984; Berger and Colella, 1989). However, this process can be quite expensive, which is why mesh-free methods have become an attractive option that goes around these limitations.

## 3.2 MAgNet: Mesh-Agnostic Neural PDE Solver

### 3.2.1 "Encode-Interpolate-Forecast" framework

Let $\{x_1, x_2, \ldots, x_T\} \in \mathbb{R}^{C \times N}$ denote a sequence of $T$ frames that represents the ground-truth data coming from a PDE simulator or real-world observations. $C$ denotes the number of physical channels, that is the number of physical variables involved in the PDE and $N$ is the number of points in the mesh. These frames are defined on the *same* mesh, that is the mesh does not change in time. We call that mesh the *parent mesh* and denote its normalized coordinates of dimensionality $n$ by $\{p_i\}_{1 \leq i \leq N} \in [-1, 1]^n$. Let $\{c_i\}_{1 \leq i \leq M} \in [-1, 1]^n$ denote a set of $M$ coordinates representing the spatial queries. The task is to predict the solution for subsequent time steps both at: (i) all coordinates from the parent mesh $\{p_i\}_{1 \leq i \leq N}$, and (ii) coordinates from the spatial queries $\{c_i\}_{1 \leq i \leq M}$. At test time, the model can be queried at any spatially continuous coordinate within the PDE domain to provide an estimate of the PDE solution at those coordinates.

To perform the prediction, we first estimate the PDE solutions at the spatial queries for the first $T$ frames and then use that to forecast the PDE solutions at the subsequent timesteps at the query locations. We do this through three stages (see Figure 1):

1. **Encoding**: The encoder takes as input the given PDE solution $\{x_t\}_{1 \leq t \leq T}$ at each point of the parent mesh $\{p_i\}_{1 \leq i \leq N}$ and generates a state-representation of original frames, which can be referred to as embeddings, and which we note $\{z_t\}_{1 \leq t \leq T}$. This representation will be used in the interpolation step to find the PDE solution at the spatial queries $\{c_i\}_{1 \leq i \leq M}$. Note that in this encoding step, we can generate one embedding for each frame such that we have $T$ embeddings or summarize all the information in the $T$ frames into one embedding. We will explain the methodology using $T$ embeddings, as it is easier to grasp the time dimension in this formulation, but the implementation has been done using a summarized

single embedding, as mentioned in section 3.2.2. We also note that the embedded mesh remains the same, i.e. we don't change it by upsampling or downsampling it.

2. **Interpolation**: We follow the same approach as Jiang et al. (2020) and Chen et al. (2020) by performing an interpolation in the feature space. Note that in case we generate one representation that summarizes all $T$ frames into one, then $z_t = z$ for $t = 1, \ldots, T$. Let $\{t_k\}_{1 \leq k \leq T}$ denote the timesteps at which the $x_t$ are generated.

   For each spatial query $c_i$, let $\mathcal{N}(c_i)$ denote the nearest points in the parent mesh $p_j$. We generate an interpolation of the features $z_k[c_i]$ at coordinates $c_i$ and at timestep $t_k$ as follows:

$$\forall k \in \{1, T\}, \forall i \in \{1, M\} : z_k[c_i] = \frac{\sum_{p_j \in \mathcal{N}(c_i)} w_j g_\theta(x_k[p_j], z_k[p_j], c_i - p_j, t_k)}{\sum_{p_j \in \mathcal{N}(c_i)} w_j} \quad (3)$$

   Where $z_k[p_j]$ and $x_k[p_j]$ denote the embedding and input frame at position $p_j$ and time $t_k$ respectively. Moreover, $w_j$ are interpolation weights and are positive and sum to one. Weights are chosen such that points closer to the spatial query have a higher contribution to the interpolated feature than points farther away from the spatial query. The $g_\theta$ is an MLP. To get the PDE solution $x_k[c_i]$ at coordinate $c_i$, we use a decoder $d_\theta$ which is an MLP here: $x_k[c_i] = d_\theta(z_k[c_i])$. In practice, the number of neighbors that we choose is $2^n$ where $n$ is the dimensionality of the coordinates.

3. **Forecasting**: Now that we generated the PDE solution at the spatial queries $c_i$ for all the past frames, we forecast the PDE solution at future time points at both spatial queries and the parent mesh coordinates. Let $\mathcal{G}$ denote the Nearest-Neighbors Graph (NNG) that has as nodes all the $N$ locations in the parent mesh (at original coordinates $\{p_i\}_{1 \leq i \leq N}$) as well as all the $M$ query points (at locations $\{c_i\}_{1 \leq i \leq M}$), with edges that include only the nearest neighbors of each node among the $N + M - 1$ others. This corresponds to a new mesh represented by the graph $\mathcal{G}$. Let $\{c'_i\}_{1 \leq i \leq M+N}$ denote the corresponding new coordinates. We generate the PDE solution for subsequent time steps on this graph auto-regressively using a decoder $\Delta_\theta$ as follows:

$$x_{k+1}[c'_i] = x_k[c'_i] + (t_{k+1} - t_k)\Delta_\theta(x_k[c'_i], \ldots, x_1[c'_i]), \quad k = T, T+1, \ldots \quad (4)$$

We train MAgNet by using two losses:

- **Interpolation Loss**: This loss makes sure that the interpolated points match the ground-truth and is computed as follows:

$$L_{\text{interpolation}} = \frac{\sum_{i=1}^{M} \sum_{k=1}^{T} ||\hat{x}_k[c_i] - x_k[c_i]||_1}{T \times M} \quad (5)$$

  Where $\hat{x}_k[c_i]$ denotes the interpolated values generated by the model at the spatial queries.

- **Forecasting Loss**: This loss makes sure that the model predictions into the future are accurate. If $H$ is the horizon of the predictions, then we can express the loss as follows:

$$L_{\text{forecasting}} = \frac{\sum_{i=1}^{M+N} \sum_{k=1}^{H} ||\hat{x}_{k+T}[c'_i] - x_{k+T}[c'_i]||_1}{H \times (M + N)} \quad (6)$$

  Where $\hat{x}_{k+T}[c'_i]$ denotes the forecasted values generated by the model at the graph $\mathcal{G}$ which combines both spatial queries and the parent mesh.

The final loss is then expressed as:

$$L = L_{\text{forecasting}} + L_{\text{interpolation}}.$$

### 3.2.2 Implementation Details

In the previous section, we described the general MAgNet framework. In this section, we present how we build the inputs to MAgNet as well as the architectural choices for the encoding, interpolation and forecasting modules and suggest two main architectures: MAgNet[CNN] and MAgNet[GNN].

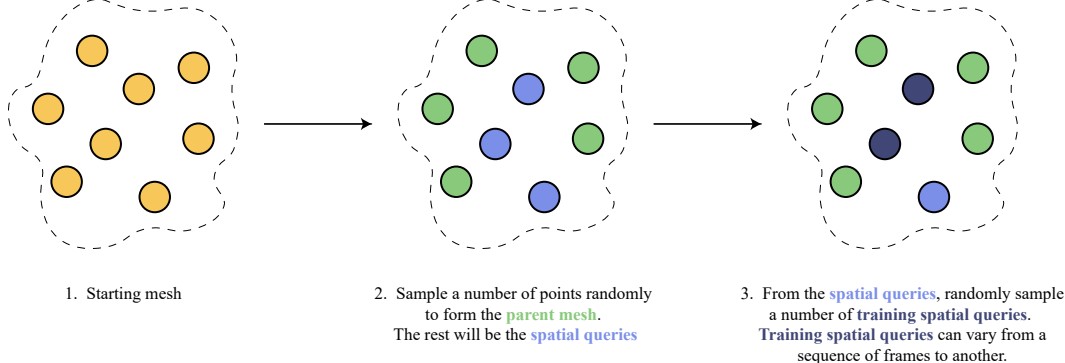

1. Starting mesh
2. Sample a number of points randomly to form the **parent mesh**. The rest will be the **spatial queries**
3. From the **spatial queries**, randomly sample a number of **training spatial queries**. **Training spatial queries** can vary from a sequence of frames to another.

Figure 2: We illustrate the data pre-processing pipeline. We sample points randomly from the **starting mesh** to form the **parent mesh** and the remaining points form the **spatial queries**. Next, during training, we can sample from the **spatial queries** and form what we call "**training spatial queries**". The distinction is that the number of "**training spatial queries**" can be less than the total number of **spatial queries** and we investigate the impact of this number in Section 4.3.

**Data pre-processing**: We first consider a mesh that contains $N'$ points ($N' \geq N$). We randomly sample $N$ points from the mesh to form the parent mesh. During training, $M$ spatial queries are randomly sampled from the $N' - N$ remaining points. We tried multiple values of $M$ (that is the number of training spatial queries) to assess its impact on the performance of the method within a sensitivity study presented in in Section 4.3. The data pre-processing is illustrated in Figure 2

**MAgNet[CNN]**: In this architecture, we follow Chen et al. (2020) and adopt the EDSR architecture (Lim et al., 2017) as our CNN encoder. We concatenate all frames $\{x_t\}_{1 \leq t \leq T}$ in the channel dimension and feed that to our encoder in order to generate a single representation $z$. For the forecasting module, we use the same GNN as in (Sanchez-Gonzalez et al., 2020). A key advantage of this architecture is that it effectively turns existing CNN architectures into mesh-agnostic ones by querying them at any spatially continuous point of the PDE domain at test time.

**MAgNet[GNN]**: This model is similar to MAgNet[CNN] except that instead of using a CNN as an encoder, we use a GNN: the same architecture as in the forecasting module but each architecture having its separate set of parameters. This is better suited for encoding frames with irregular meshes. Similarly to MAgNet[CNN], we generate a single representation $z$ that summarizes all the information from the frames $\{x_t\}_{1 \leq t \leq T}$.

## 4 Results

In this section, we evaluate MAgNet's performance against the following baselines:

**Fourier Neural Operators (FNO) (Li et al., 2021)** : Considered the state-of-the-art model in neural PDE solving, FNO casts the problem of PDE solving as learning an operator from the space of initial conditions to the space of the solutions. It is able to learn PDE solutions that lie on a uniform grid and can do zero-shot super resolution.

**Message-Passing Neural PDE Solvers (MPNN) (Brandstetter et al., 2022)** : Graph Neural Networks have been used to learn physical simulations with great success (Sanchez-Gonzalez et al., 2020). Recently, they have been used to learn solutions to PDEs (Brandstetter et al., 2022; Sanchez-Gonzalez et al., 2020). MPNN-based GNNs coupled with an autoregressive strategy demonstrate a superior performance to FNO and are able to make long rollouts with the help of the "pushforward-trick" that only propagates gradient of the last computed frame.

We evaluate all models on both 1D and 2D simulations (the datasets generated from 1D and 2D PDEs are presented in Appendix A.4). All training sets in the 1D case contain 2048 simulations and test sets contain 128 simulations. For the 2D case, training sets contain 1000 simulations and test sets contain 100 simulations. All models are evaluated using the Mean Absolute Error (MAE) on the rolled out predictions averaged across time and space:

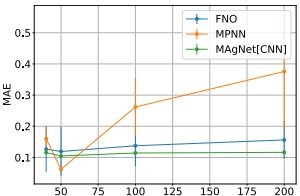 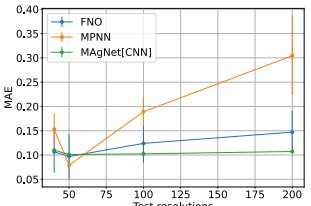 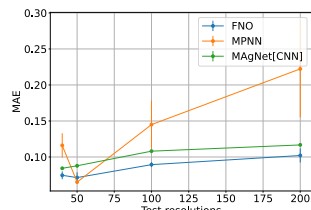

(a) Zero-shot super-resolution performance on **E1** test.

(b) Zero-shot super-resolution performance on **E2** test.

(c) Zero-shot super-resolution performance on **E3** test.

Figure 3: We present the models predictive performance on zero-shot super-resolution, with a training spatial resolution of $n_x = 50$ for the regular grids. MAgNet[CNN] outperforms baselines on both **E1** and **E2** test sets but lags behind FNO on **E3**. Error bars represent one standard deviation in both plots.

$$MAE = \frac{\sum_{t=1}^{n_t-T} \sum_{i=1}^{N} |x_t[c_i] - \hat{x}_t[c_i]|}{(n_t - T) \times N}. \tag{7}$$

Where $n_t$ is the total number of frames and $T$ the number of frames input to the models.

We train models for 250 epochs with early stopping with a patience of 40 epochs. See Appendix A.6 and A.7 for more implementation details.

We present a short summary of our results:

- **1D Case**: Our model MAgNet is able to robustly generalize to unseen meshes in the regular case as compared to the SOTA models FNO and MPNN. The performance is even better in the irregular case. The details of the results are given in the following section 4.1.

- **2D Case**: Our model MAgNet is less competitive than MPNN when it comes to generalizing to unseen meshes in the regular case. However, in the irregular case, we are more competitive especially when trained on sparse meshes. The results regarding the 2D PDEs are presented in section 4.2.

## 4.1   1D Case

For all the subsequent sections, $n_x$ and $n'_x$ denote the training and testing set's resolutions respectively. The temporal resolution $n_t = 250$ remains unchanged for all experiments.

### 4.1.1   General Performance and zero-shot super resolution On Regular Meshes

In this section, we compare MAgNet's performance on all three datasets. All models are trained on a resolution of $n_x = 50$ and the PDE solutions lie on a uniform grid. We test zero-shot super-resolution on $n'_x \in \{40, 50, 100, 200\}$. Results are summarized in Figure 3 and visualizations of the predictions can be found in Appendix A.2. MAgNet[CNN] outperforms both baselines on both **E1** and **E2** datasets yet is slightly outperformed by FNO on **E3** (Figure 3c). Nonetheless, MAgNet[CNN]'s predictive performance stays consistent up to $n'_x = 200$ while MPNN does not generalize well to resolutions not seen during training.

### 4.1.2   General Performance and zero-shot super-resolution on Irregular Meshes

In this section, we study how MAgNets compare against the other baselines when it comes to making predictions on irregular meshes. In order to do so, we take simulations from the uniform-mesh **E1** dataset with a resolution of 100 do the following:

Let $n_x \in \{30, 50, 70\}$. Then, for each simulation in the **E1** dataset, randomly sample the same subset of $n_x$ points: the mesh remains for each single simulation in the **E1** dataset.

The procedure is the same for the test set but we instead take the original **E1** test set at a starting resolution of 200 and generate four test sets with irregular meshes for $n'_x \in \{40, 50, 100, 200\}$. This

| | $n_x = 30$ | | | | $n_x = 50$ | | | | $n_x = 70$ | | | |
|---|---|---|---|---|---|---|---|---|---|---|---|---|
| **Model** | $n'_x = 40$ | $n'_x = 50$ | $n'_x = 100$ | $n'_x = 200$ | $n'_x = 40$ | $n'_x = 50$ | $n'_x = 100$ | $n'_x = 200$ | $n'_x = 40$ | $n'_x = 50$ | $n'_x = 100$ | $n'_x = 200$ |
| **FNO** | 0.2784 | 0.2471 | 0.2574 | 0.2501 | 0.3797 | 0.3324 | 0.3841 | 0.3821 | 0.2798 | 0.2341 | 0.2533 | 0.2605 |
| **MAgNet[CNN]** | **0.2081** | 0.1934 | 0.2063 | 0.2150 | 0.1869 | **0.1630** | **0.1599** | 0.1629 | **0.2237** | 0.1634 | 0.1385 | 0.1324 |
| **MPNN** | 0.2602 | **0.1601** | 0.3451 | 0.3667 | 0.3027 | 0.2521 | 0.3226 | 0.3243 | 0.2685 | **0.1541** | 0.3403 | 0.3570 |
| **MAgNet[GNN]** | 0.2422 | 0.2230 | **0.1938** | **0.1902** | 0.2302 | **0.1659** | **0.1590** | **0.1404** | 0.2400 | **0.1599** | **0.1398** | **0.1070** |

Table 1: We report the MAE per frame on the **E1** dataset. We train all four models on three different resolutions $n_x \in \{30, 50, 70\}$ and for each training resolution, we evaluate zero-shot super-resolution on irregular meshes for $n'_x \in \{40, 50, 100, 200\}$. We notice that even when we use a CNN encoder, MAgNet not only performs better than the existing baselines, but its performance stays consistent across different test resolutions. MAgNet with a CNN encoder beats MPNN even when using an encoder not suited for the task, which suggests MAgNet successfully turns existing CNN architectures into mesh-agnostic ones.

is different from the test set of the previous section albeit considering the same resolution since this one has irregular meshes. We summarize our findings in Table 1.

MAgNet[GNN] performs better than MAgNet[CNN] on irregular meshes which is expected since GNN encoders are better suited for this task. However, surprisingly, even though we use a CNN encoder for MAgNet[CNN], the performance seems to be better in most cases not only compared to FNO but also MPNN which is a graph-based architecture. This effectively shows that MAgNet can be used to turn existing CNN architectures into mesh-agnostic solvers. This is particularly interesting for meteorological applications where one needs to make predictions at the sub-grid level (at a specific coordinate) while only having access to measurements on a grid.

## 4.2 2D case

In this section, we present results for the 2D PDE simulations. We use the datasets **B1** and **B2** as our experimental testbeds (see appendix A.4). We use $n_{train}$ to denote the train resolution and $n_{test}$ for the test resolution. All models are fed a history of $T = 10$ frames and are required to generate a rollout of $n_t - T = 40$ frames in the future.

### 4.2.1 General performance and zero-shot super resolution on regular meshes

In this section, we compare MAgNet's performance on both **B1** and **B2** datasets. All models were trained on a resolution of $n_{train} = 64^2$ and the PDE solutions lie on a uniform grid. Zero-shot super resolution is tested on $n_{test} \in \{32^2, 64^2, 128^2, 256^2\}$. We summarize our findings in Figure 4. We notice that MAgNet[CNN] falls behind when it comes to making good predictions and zero-shot super-resolution on both datasets while MAgNet[GNN] and MPNN take the lead. Suprisingly, FNO suffers when it comes to generalizing to unseen resolutions. Indeed, leveraging interactions between points after the interpolation module allows MAgNet[GNN] not only to make good predictions but also to generalize to denser and unseen resolutions.

Visualizations of the prediction for a sample for the **B1** dataset for all test resolutions can be found in Figures 10, 11, 12 and 13 respectively in the appendix.

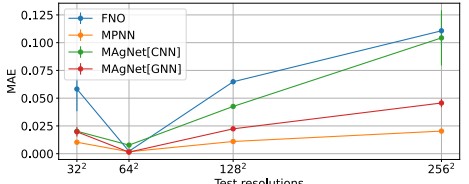
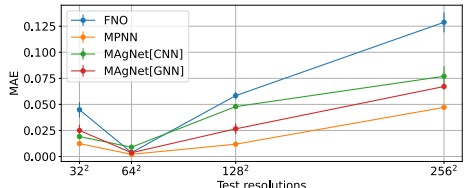

(a) Results for the zero-shot super-resolution on the **B1** dataset.

(b) Results for the zero-shot super-resolution on the **B2** dataset.

Figure 4: Results for the zero-shot super-resolution setting for models trained in the regular setting on a resolution of $n_{train} = 64^2$. While FNO and MAgNet[CNN] fall behind, MPNN and MAgNet[GNN] take the lead by leveraging the message-passing paradigm of Graph Neural Networks (Brandstetter et al., 2022; Gilmer et al., 2017)

#### 4.2.2 General performance and zero-shot super resolution on irregular meshes

In this section, we present results for irregular meshes. We consider two settings:

- Uniform: $N$ nodes are randomly and uniformly sampled from a regular grid of resolution $32^2$

- Condensed: $N$ nodes are randomly sampled in a non-uniform way from a regular grid of resolution $32^2$ following the distribution: $p(x, y) \propto \exp\left(-8\left((x - 0.25)^2 + (y - 0.25)^2\right)\right)$

We train MPNN and MAgNet[GNN] on four resolutions $N \in \{64, 128, 256, 512\}$ where again $N$ is the number of nodes in the mesh for both Uniform and Condensed settings. For each training setting, we evaluate both models on regular grid PDE simulations from the **B1** dataset. Figure 18 shows the mesh nodes for the Uniform setting and Figure 19 is for the Condensed setting.

Our findings are summarized in Figure 5. We notice that our model MAgNet[GNN] is especially more appealing in case we have fewer points to work with. In the Condensed setting which is more realistic than the uniform one, MAgNet[GNN] is even more competitive than MPNN. We note however, that while MAgNet[GNN]'s performance is better for small test resolutions, it quickly deteriorates as we test on higher resolutions which is a main limitation of our approach in the 2D case.

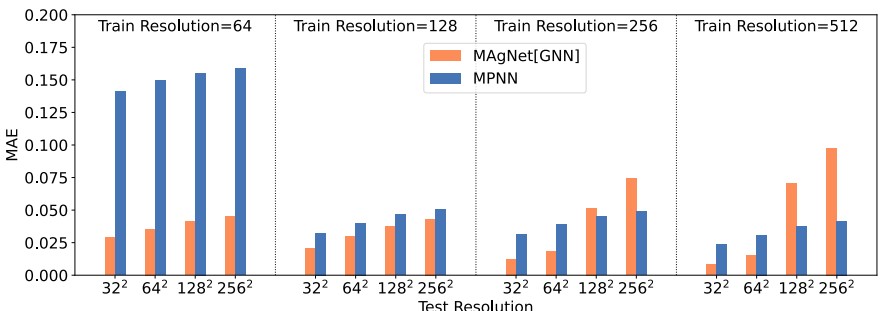

(a) Results for the zero-shot super-resolution on the irregular **B1** dataset for the Uniform setting.

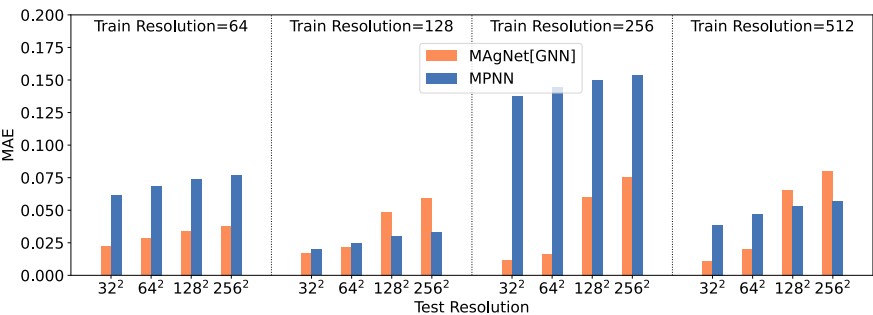

(b) Results for the zero-shot super-resolution on the irregular **B1** dataset for the Condensed setting.

Figure 5: Results for the zero-shot super-resolution setting for models trained in the irregular setting on resolutions of $N \in \{64, 128, 256, 512\}$. MAgNet[GNN] is especially appealing in case we have fewer points and in case the node distribution is non-uniform. However, the performance gap reduces as both models see more points during training. We note however, that MAgNet[GNN]'s performance deteriorates as we test on higher resolutions compared to MPNN which is a main limitation of our approach in the 2D case.

### 4.3 Ablation study: Basic Interpolators vs Learned Interpolators

We investigate the contribution of the interpolation module to the general predictive performance of MAgNet. We compare the MAgNet[CNN] architecture against three ablated variants:

- **KNN**: We use K-Nearest-Neighbors interpolation (Qi et al., 2017) on the original frames directly to obtain the interpolated values at the spatial queries.

- **Linear**: We use Linear interpolation on the original frames directly to obtain the interpolated values at the spatial queries.

- **Cubic**: We use Cubic interpolation on the original frames directly to obtain the interpolated values at the spatial queries.

Everything else is kept the same. The evaluation is done on the **E1** dataset with regular meshes and a resolution of $n_x = 50$. Performance is tested on **E1** with regular meshes for test resolutions $n'_x \in \{40, 50, 100, 200\}$. Results are summarized in Figure 6 and additional ablation studies can be found in the appendix A.1.

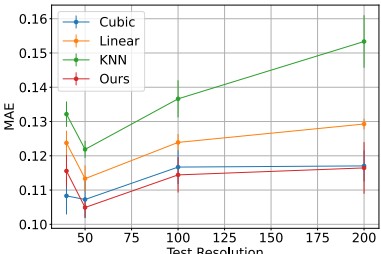

Figure 6: We study the impact of having a learned interpolator (Ours) as compared to existing interpolation schemes. Error bars represent one standard deviation.

## 5 Limitations and Future Work

In this paper we introduced a novel framework that we call MAgNet for solving PDEs on any mesh, possibly irregular. We proposed two variants of the architecture which gave promising results on benchmark datasets. We were effectively able to beat graph-based and grid-based architectures even when using the CNN variant of the proposed framework, therefore suggesting a novel way of adapting existing CNN architectures to make predictions on any mesh. The main added value of the proposed method is its very good performance on irregular meshes, including for super-resolution tasks, as it can be observed for the presented 1D and 2D experiments, when compared to SOTA methods. Notably, it seems to perform best when handed the smallest amount of data to work with (with a 64x64 training resolution), even in the condensed non-uniform case. This is a very desirable property for real-life data. A limitation of our work however, is the significance of the learned interpolator. Indeed, compared with a simple cubic interpolation, the approach introduced here doesn't seem to offer a significant advantage and we leave improvement regarding this point for future work. Another improvement could be seen in the forecasting module. For now, MAgNet forecasts using a first-order explicit time-stepping scheme that is known to suffer from instability problems in numerical PDE and ODE solvers. Learned solvers seem to somehow circumvent this limitation even when using large time steps (Sanchez-Gonzalez et al., 2020; Brandstetter et al., 2022; Stachenfeld et al., 2021). In a future work, we wish to explore other time-stepping schemes such as the 4th order Runge-Kutta method (Runge, 1895; Kutta, 1901) which is commonly used for solving PDEs. Finally, we observed that our MAgNet model appears to perform significantly better in a 1D than in a 2D setting (as shown in section 4.2). While it is retaining its superiority in an irregular mesh setting, it appears a little less performant than the MPNN method, even with its GNN variant. Further research on why the regular setting poses issues is thus also reserved for future work.

## Acknowledgments and Disclosure of Funding

This work is financially supported by the government of Quebec and Samsung. The authors would like to thank Shruti Mishra, Victor Schmidt, Dianbo Liu, and Ayoub Ajarra for their fruitful discussions and useful insights.

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
