# A   Appendix

## A.1   Additional Ablation and sensitivity studies

In this section, we study different architectural choices and the sensitivity of key parameters in MAgNet.

**Effect of modeling interactions between the parent mesh and the spatial query**   In section 3.2, we presented the developed method which can be used to generate solutions at any spatial query through the "Encode-Interpolate-Forecast" framework. This means that we are free to choose any architecture for the three processes. In this section we investigate the choice of the "Forecast" architecture on the predictive performance as well as zero-shot super-resolution capabilities. We compare MAgNet[CNN] and a variant that uses LSTM with attention ((Hochreiter and Schmidhuber, 1997; Bahdanau et al., 2015) on the spatial queries only. Results are shown in Table 2. We see that leveraging the interaction between the coordinates from the spatial query and those in the parent mesh enables the model to give predictions consistent to different resolutions as opposed to generating the solutions at these queries with no interaction.

Table 2: Mean Absolute Error (MAE) reported for models trained on the **E1** dataset with a resolution of $n_x = 50$ on a uniform grid and tested on the same dataset for $n_x \in \{40, 50, 100, 200\}$. We evaluate the effect of the Forecast module on the zero-shot super-resolution capabilities. The model without interaction contains an LSTM with attention (Bahdanau et al., 2015; Hochreiter and Schmidhuber, 1997) for forecasting where spatial queries do not interact with the parent mesh. The model with interaction has a GNN that operates over the graph formed from the spatial-queries and the parent mesh (MAgNet[CNN]).

| Model | $n_x = 40$ | $n_x = 50$ | $n_x = 100$ | $n_x = 200$ |
|---|---|---|---|---|
| **Without Interaction** | 0.1650 | 0.0815 | 0.2810 | 0.4139 |
| **With Interaction** | 0.1079 | 0.1020 | 0.1142 | 0.1177 |

**Impact of the number of point samples during training**   We study the impact of the number of spatial queries used during training ($M$). We train MAgNet[GNN] on the **E1** dataset with a resolution of $n_x = 50$ on a uniform grid and test on the same resolution. Our findings are summarized in Figure7. Increasing the number of spatial queries increases the predictive performance as expected. Moreover, having many queries also decreases the variance of the results. When we have fewer points, the random sampling can cause some of these points to be in regions that decrease the loss faster than other regions, hence, the model's performance becomes sensitive to randomization. However, this effect grows weaker as the number of queries increases since they would uniformly cover more regions in the mesh.

## A.2   Visualizations: Models' predictions in the 1D case

Figures 8 and 9 show the 1D models' predictions on each of the test set resolutions $n'_x \in \{40, 50, 100, 200\}$ on the **E1** dataset. MAgNet[CNN] predictions visually match the ground-truth's while MPNN's prediction degrade as the predictions are advanced in time. The models shown here are the ones trained on uniform meshes.

## A.3   Visualizations: Models' predictions in the 2D case

### A.3.1   Regular case

Figures 10, 11, 12 and 13 show the 2D models' predictions on each of the methods on the **B1** dataset, trained on regular meshes, for four different test resolutions. The models are all trained on a 64×64 grid. Visually MPNN and MAgNet[GNN] predictions appear rather close, and both matching relatively well the ground-truth's.

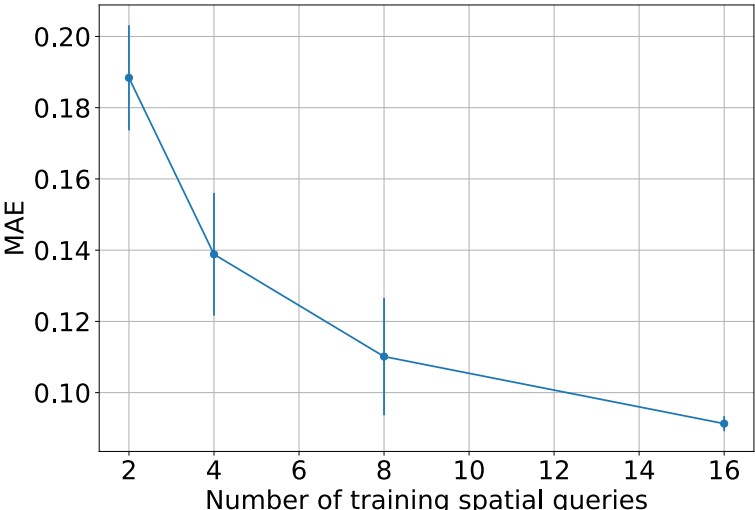

Figure 7: We assess the impact of the number of spatial queries during training. Error bars represent one standard deviation.

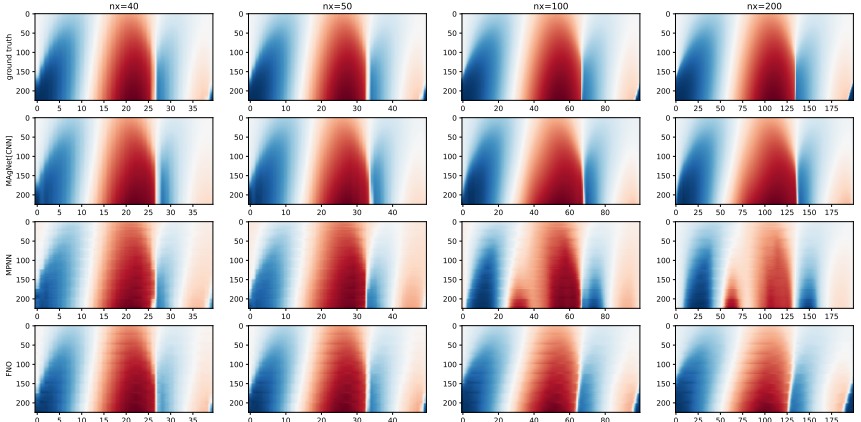

Figure 8: Vizualisation of the models' 1D predictions on a simulation sample from the **E1** dataset. We present visualizations for each of the test resolutions $n'_x \in \{40, 50, 100, 200\}$. The temporal resolution is fixed at $n_t = 250$. The x axis represents space and y axis represents time. The arrow of time is from top to bottom.

### A.3.2 Irregular case

Figures 14, 15, 16 and 17 show the 2D models' predictions on the **B1** dataset for the Uniform setting on resolutions $n_{test} \in \{32^2, 64^2, 128^2, 256^2\}$ respectively (see Section 4.2.2):

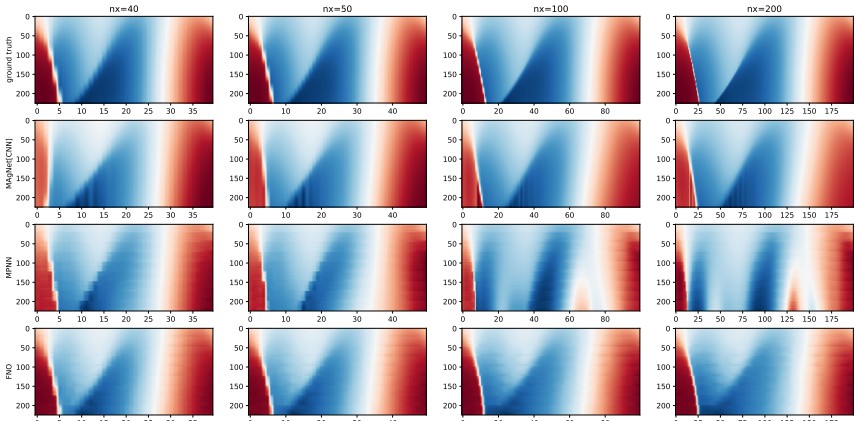

Figure 9: Vizualisation of the models' 1D predictions on a simulation sample from the **E1** dataset. We present visualizations for each of the test resolutions $n'_x \in \{40, 50, 100, 200\}$. The temporal resolution is fixed at $n_t = 250$. The x axis represents space and y axis represents time. The arrow of time is from top to bottom.

## A.4 PDE Datasets

**1D PDE Simulations**    For the 1D case, We use three of MPNN's PDE simulations (Brandstetter et al., 2022) as our experimental testbed. In the same fashion, we are interested in the following family of PDEs:

$$\begin{cases} [\partial_t u + \partial_x(\alpha u^2 - \beta \partial_x u + \gamma \partial_{xx} u)](t, x) = \delta(t, x) \\ u(0, x) = \delta(0, x) \\ \delta(t, x) = \sum_{j=1}^{J} A_j \sin(\omega_j t + 2\pi l_j x/L + \phi_j) \end{cases} \tag{8}$$

Where $J = 5$, $L = 16$ and coefficients sampled uniformly in $A_j \in [-0.5, 0.5]$, $\omega_j \in [-.4, -0.4]$, $l_j \in \{1, 2, 3\}$, $\phi_j \in [0, 2\pi)$ and periodic boundary conditions following Brandstetter et al. (2022); Bar-Sinai et al. (2018). For the 1D case, the temporal resolution is set to $n_t = 250$ for the entire study. During testing, all the models are fed a history of $T = 25$ frames and produce a rollout of $n_t - T = 225$ frames in the future. Here, we present the two datasets we work with:

- **E1**: Burgers equation without diffusion $\alpha = 1, \beta = 0, \gamma = 0$
- **E2**: Burgers equation with variable diffusion $\alpha = 1, \beta = \eta, \gamma = 0$ where $\eta \in [0, 0.2]$
- **E3**: Mixed scenario where $\alpha \in [0, 3], \beta \in [0, 0.4]$ and $\gamma \in [0, 1]$.

**2D PDE Simulations**    For the 2D case, we use the Burgers equation as our experimental testbed. This dataset was generated using Phiflow [3]:

$$\begin{cases} [\partial_t u + u\nabla u](t, x, y) = \beta[\Delta u](t, x, y) \\ u(0, x, y) = f(x, y) \\ f(x, y) = \sum_{j=1}^{J} A_j \sin(2\pi l_j^x x/L + \phi_j^x) \cos(2\pi l_j^y y/L + \phi_j^y) \end{cases} \tag{9}$$

Where $J = 5$, $L = 64$ and coefficients sampled uniformly in $A_j \in [-0.5, 0.5]$, $l_j^x, l_j^y \in \{1, 2, 3\}$, $\phi_j^x, \phi_j^y \in [0, 2\pi)$ and periodic boundary conditions. For the 2D case, the temporal resolution is set to $n_t = 50$ for the entire study. During testing, all the models are fed a history of $T = 10$ frames and produce a rollout of $n_t - T = 40$ frames in the future. Here, we present the three datasets we work with:

---

[3]`https://github.com/tum-pbs/PhiFlow`

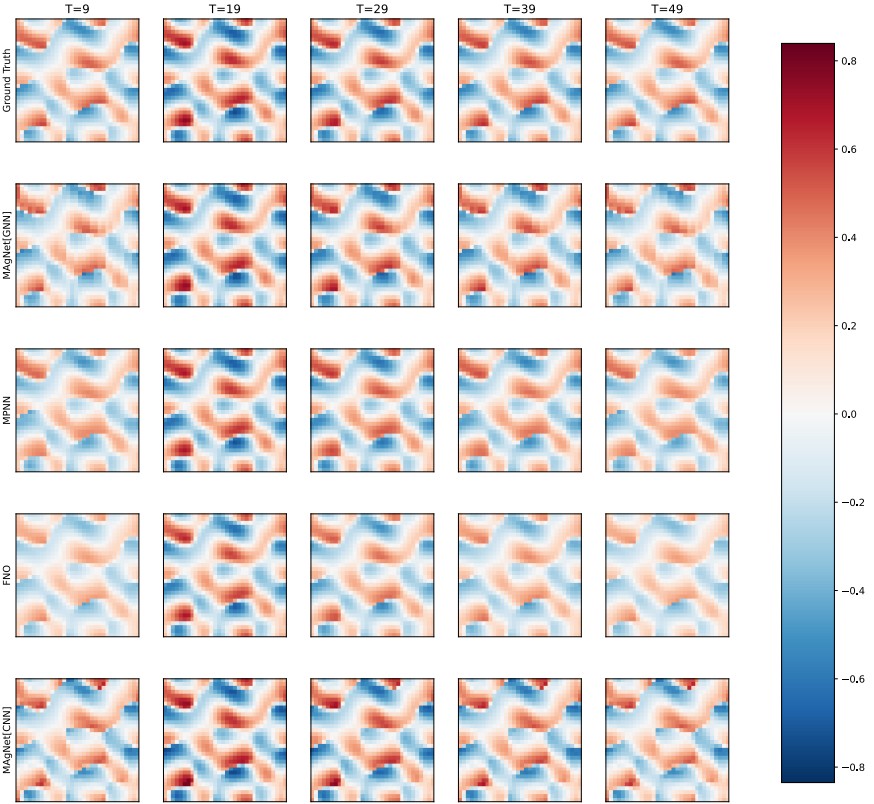

Figure 10: Visualization of the models' 2D predictions on a simulation sample from the **B1** dataset. The models are trained on a 64×64 grid, and tested on a 32×32 resolution. The shown time steps are specified on top of the figure.

- **B1**: Burgers equation without diffusion $\beta = 0$

- **B2**: Burgers equation with variable diffusion $\beta \in (0, 0.2]$

### A.5  Visualization of the irregular meshes in the 2D case

### A.6  Training details

We train all models for 250 epochs and early stopping with a patience of 40 epochs. All models are trained using Adam Optimizer (Kingma and Ba, 2014) and the StepLR learning scheduler (Paszke et al., 2019) which decays the learning rate by a factor $k$ every $N_{steps}$ epochs. All models were trained on 5 random seeds $\in \{5, 10, 21, 42, 2022\}$. We use Pytorch (Paszke et al., 2019) and Pytorch-Lightning (Falcon et al., 2020) for our implementations and we summarize all model's training hyper-parameters in Table 3.

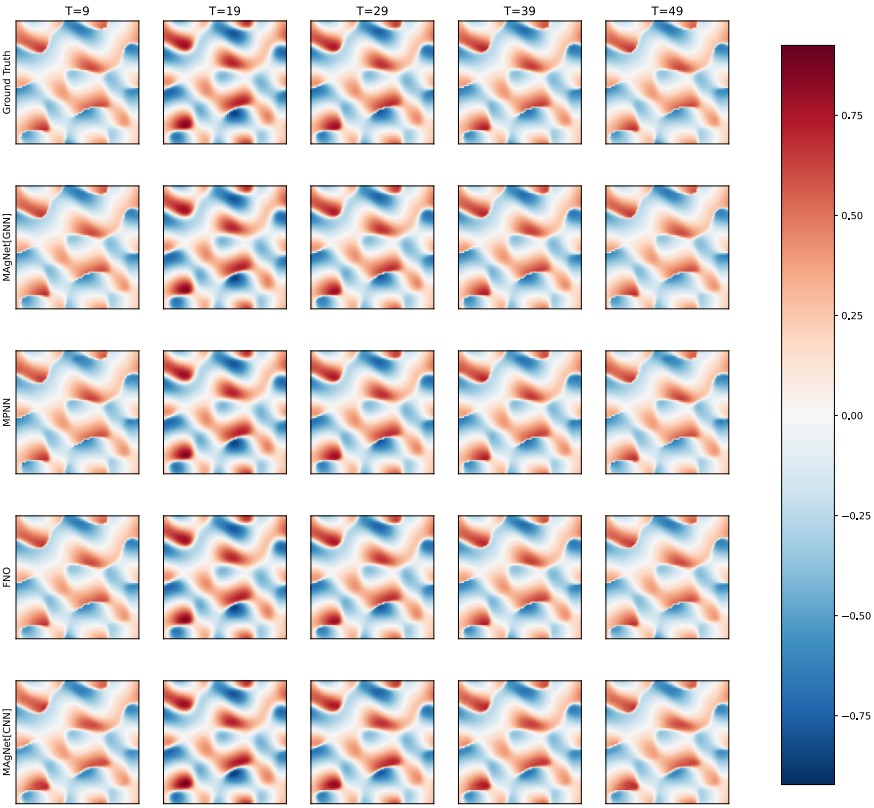

Figure 11: Visualization of the models' 2D predictions on a simulation sample from the **B1** dataset. The models are trained on a 64×64 grid, and tested on a 64×64 resolution. The shown time steps are specified on top of the figure.

## A.7 Architectural details

### A.7.1 MAgNet[CNN]

**Encoder Architecture**  We adapt the original EDSR (Lim et al., 2017) architecture to work on 1D signals instead of 2D and use 4 residual blocks with a hidden dimension of 128.

**Interpolation Module**  We use a 4 layers MLP with a hidden size of 64 followed by Layernorm for $g_\theta$. For $d_\theta$, we use a 4 layer MLP with a hidden size of 64

**Forecasting Module**  We use the same architecture as in Sanchez-Gonzalez et al. (2020). The encoder module uses a 4 layer MLP with a hidden size of 64 and the latent dimension to 32. We use 5 message-passing steps (with the same parameters for the MLP), the decoder also has a 4 layer MLP with hidden size of 64.

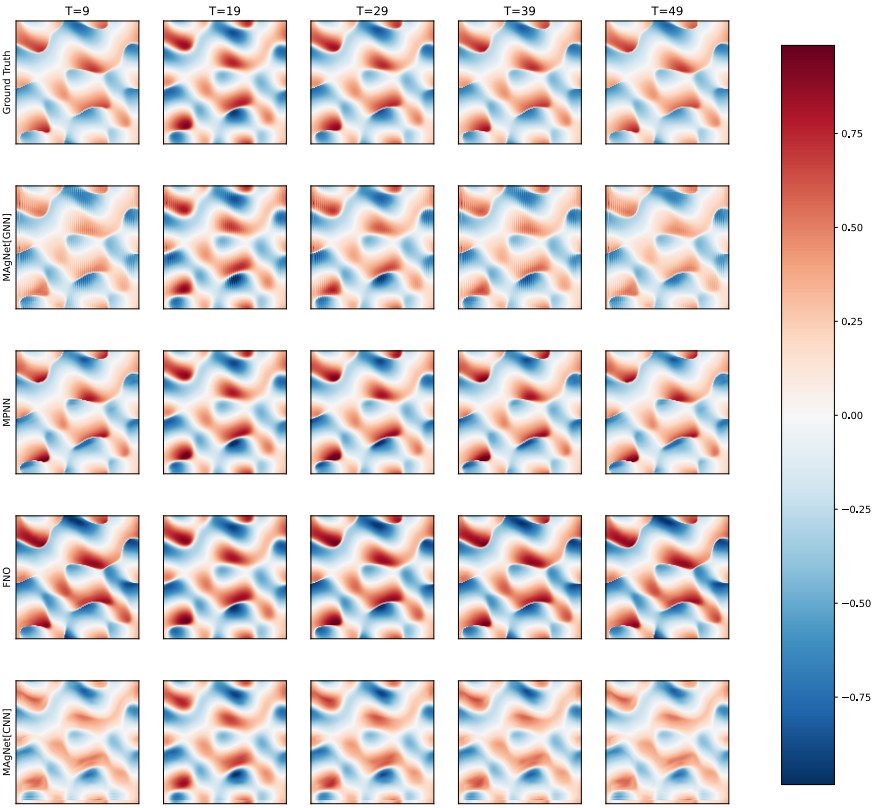

Figure 12: Visualization of the models' 2D predictions on a simulation sample from the **B1** dataset. The models are trained on a 64×64 grid, and tested on a 128×128 resolution. The shown time steps are specified on top of the figure.

### A.7.2   MAgNet[GNN]

**Encoder Architecture**   We use the same architecture as in Sanchez-Gonzalez et al. (2020) but only keep the encoder and processor. We also use 5 message-passing steps for the processor and use 4 layer MLP with hidden size of 128 and a latent dimension of 128.

**Interpolation Module**   We use a 4 layers MLP with a hidden size of 128 followed by Layernorm for $g_\theta$. For $d_\theta$, we use a 4 layer MLP with a hidden size of 128

**Forecasting Module**   We use the same architecture as in Sanchez-Gonzalez et al. (2020). The encoder module uses a 4 layer MLP with a hidden size of 64 and the latent dimension set to 128. We use 5 message-passing steps for the processor (with the same parameters for the MLP), the decoder also has a 4 layer MLP with hidden size of 128.

### A.7.3   MPNN

We use the same hyperparameters as in Brandstetter et al. (2022).

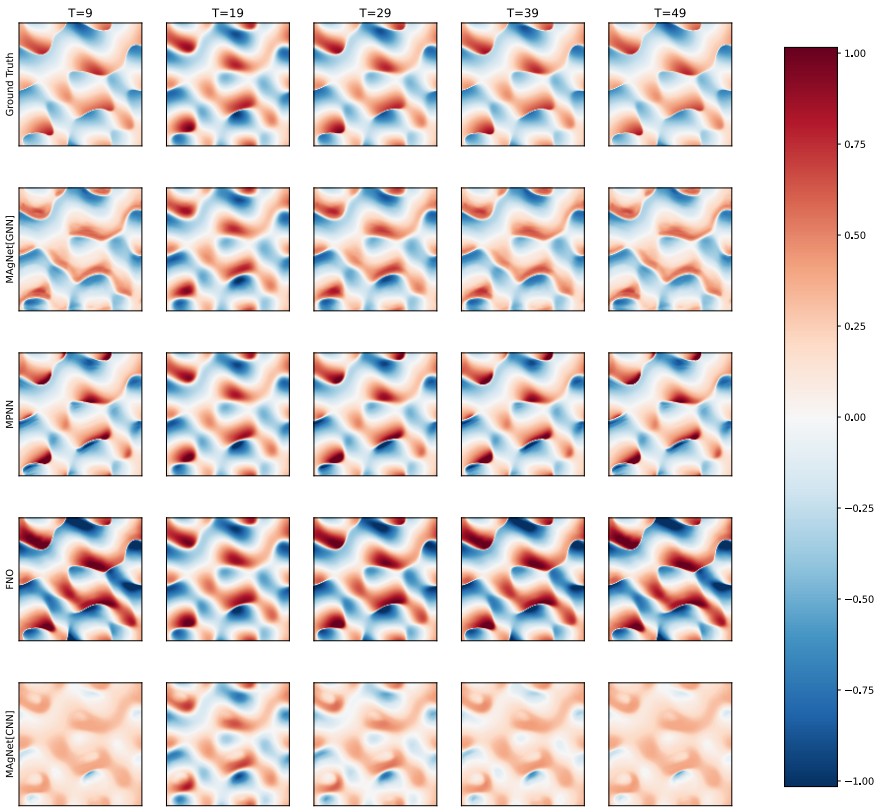

Figure 13: Visualization of the models' 2D predictions on a simulation sample from the **B1** dataset. The models are trained on a 64×64 grid, and tested on a 256×256 resolution. The shown time steps are specified on top of the figure.

### A.7.4 FNO

We use 5 Fourier Layers and 12 modes in Fourier space. We use a hidden channel size of 256.

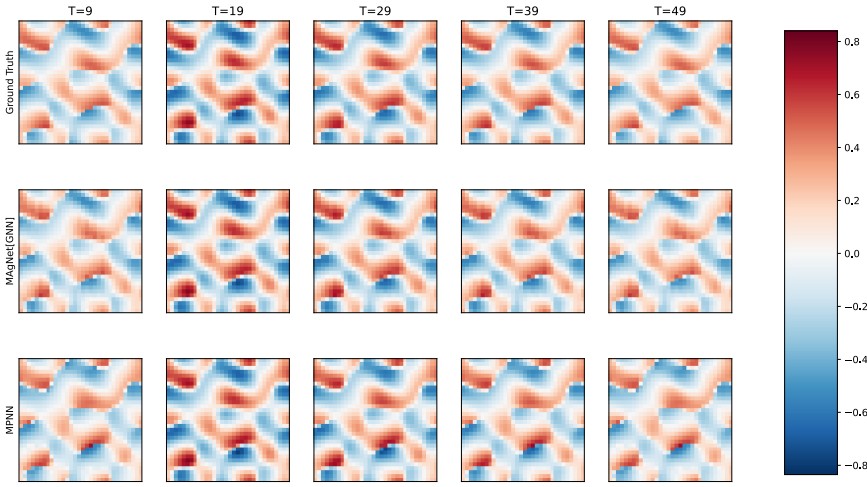

Figure 14: Visualization of the models' 2D predictions on a simulation sample from the **B1** dataset, on an irregular mesh. The models are trained on a $N = 512$ nodes, and tested on a $32\times32$ resolution. The shown time steps are specified on top of the figure.

Table 3: Training hyperparameters for FNO, MPNN, MAgNet[CNN] and MAgNet[GNN]

| Parameters | FNO | MPNN | MAgNet[CNN] | MAgNet[GNN] |
|---|---|---|---|---|
| **Learning Rate** | 0.001 | 0.001 | 0.001 | 0.001 |
| **Weight Decay** | 0 | 0 | 0 | 0 |
| $k$ | 0.3 | 0.3 | 0.3 | 0.3 |
| $N_{steps}$ | 50 | 50 | 40 | 50 |
| **GPU** | RTX8000 | RTX8000 | RTX8000 | RTX8000 |
| **Number of GPUs** | 1 | 1 | 2 | 2 |
| **Training duration (hours)** | 1 | 1 | 5 | 2.5 |

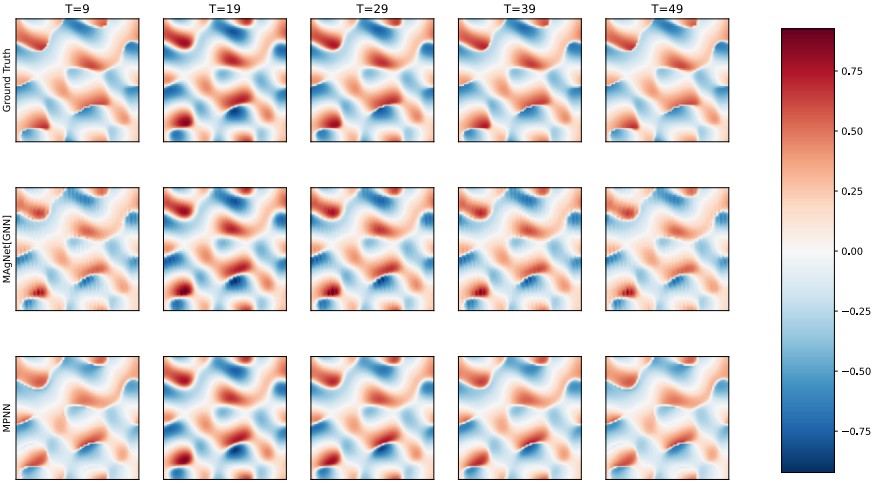

Figure 15: Visualization of the models' 2D predictions on a simulation sample from the **B1** dataset, on an irregular mesh. The models are trained on a $N = 512$ nodes, and tested on a $64 \times 64$ resolution. The shown time steps are specified on top of the figure.

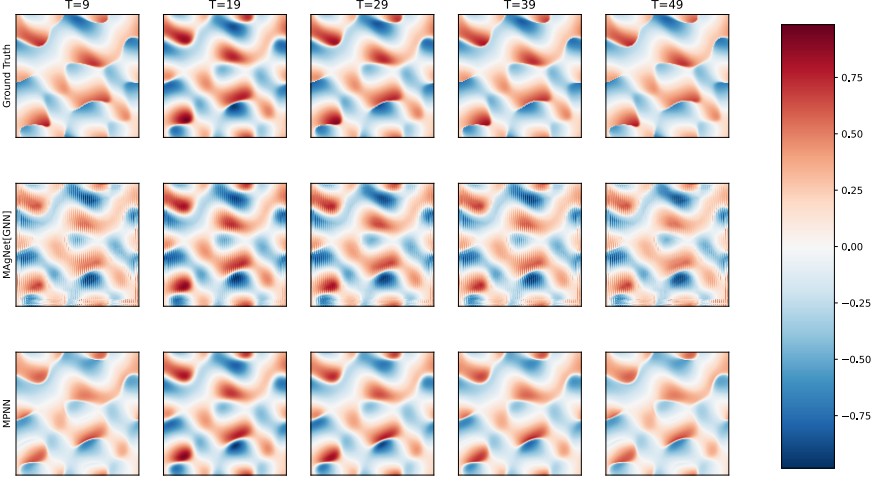

Figure 16: Visualization of the models' 2D predictions on a simulation sample from the **B1** dataset, on an irregular mesh. The models are trained on a $N = 512$ nodes, and tested on a $128 \times 128$ resolution. The shown time steps are specified on top of the figure.

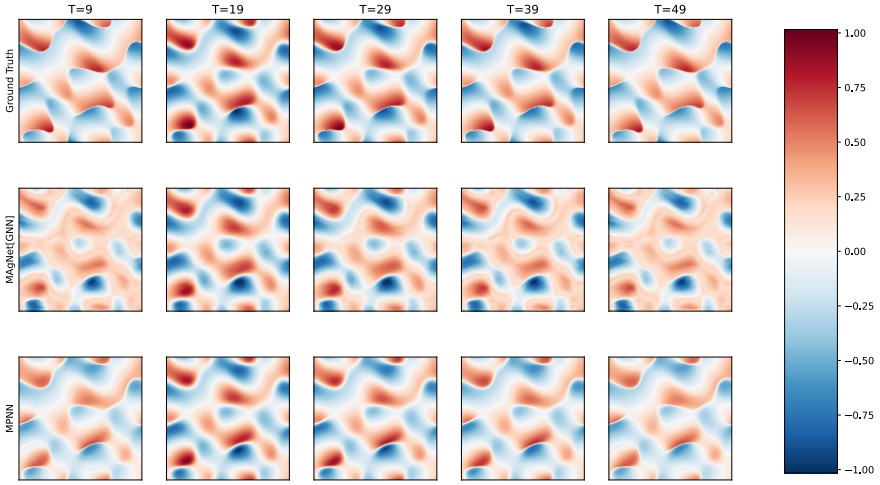

Figure 17: Visualization of the models' 2D predictions on a simulation sample from the **B1** dataset, on an irregular mesh. The models are trained on a $N = 512$ nodes, and tested on a $256{\times}256$ resolution. The shown time steps are specified on top of the figure.

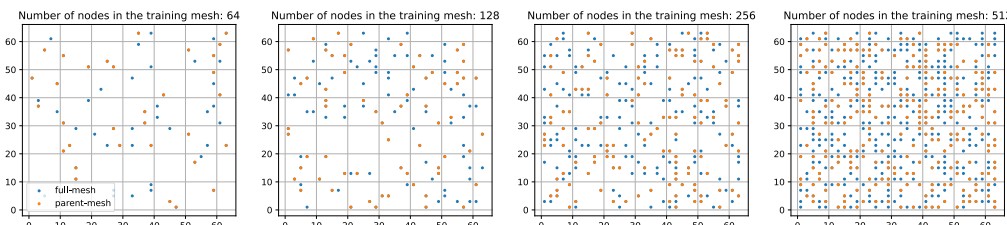

Figure 18: Visualization of an irregular mesh, uniformly distributed.

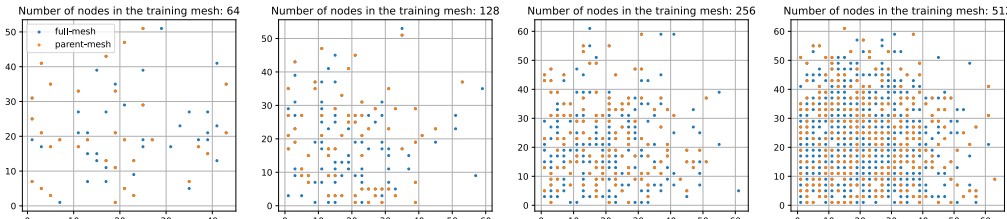

Figure 19: Visualization of an irregular mesh, non-uniformly distributed but following a bi-variate normal distribution.