# OpenReview forum: "MAgNet: Mesh Agnostic Neural PDE Solver"
_NeurIPS.cc/2022/Conference — NeurIPS 2022 Accept_

### Official Review · Reviewer_yyq7 · 2022-07-12

**Rating:** 6
**Confidence:** 3
**Soundness:** 3 good
**Presentation:** 2 fair
**Contribution:** 3 good

**Summary:**

1- Summary

The authors propose a PDE solver method based on implicit neural representations address the general limitation of generic numerical PDE solvers that assume the the underlying PDE is discretized and solve it with the Finite Element Method. More specifically, the proposed method lifts such assumptions while increasing prediction accuracy and resolution over generic numerical PDE solvers. The proposed method uses a graph neural network to infer the next state of the system in addition to a physics-based inductive bias in its loss term to constraint the neural network behavior by encouraging it to produce more physically-accurate results. The experiments show that the proposed method is able to estimate spatially continuous solution of the underlying PDE up to 250 frames with physically-consistent accuracy.

**Questions:**

- It is not clear how/when the proposed model fails. Have the authors tested their model's limits?

**Ethics Review Area:**

["I don’t know"]

**Limitations:**

I am personally hesitant to question societal impacts of any scientific discovery that is not purposefully aiming to promote certain views/directions that directly and objectively harm the society. This is because I believe in the long-term this limits the natural evolution of discovery of the Truth, similar to how protection of the free expression principle leads to democracy. Arguably, any genuine scientific discovery could be eventually used for negative purposes. Therefore, I do not think the reviewers’ subjective opinion should be used in any way as an objective to decide the direction of scientific discoveries.

Regardless, I do not think the proposed method would have any direct negative impact in the society.

**Strengths And Weaknesses:**

2- Paper strengths

- The proposed PDE solver can generalize to PDEs with higher resolution that PDEs than it was trained on
- The experiments show that the proposed method performs better or similar than SOTA methods

3- Paper weaknesses
Note that some of the weaknesses below are not meant to suggest that the authors should address them during the discussion prior in order to change my opinion. This is because I understand that addressing some of the weaknesses might take a long time or might require significant changes to the proposed method. Although addressing any of these might change my judgment about the work but for the time being the authors may prioritize addressing questions.

- The authors start the paper by talking about climate forecasting as a challenging, high-resolution, problem but their model is far from handling anything related to weather forecasting and their experiments also does not include anything for climate forecasting

4- Additional Comments

Related Work:
- The related work section is written a bit mechanistically and does not highlight some of the distinctive differences of the prior methods in contrast to the authors’ proposed method.

Results:
- It would be good if the authors could provide a summary of the results at the beginning of this section.

---

> ### Author Response · Authors · 2022-08-02
> **Official Response to Reviewer yyq7**
>
>
> We thank the reviewer for the thoughtful feedback and suggestions to improve the paper.
> We respond to the reviewer’s questions/comments below, indicating a change in the paper when it applies.
>
> >*The authors start the paper by talking about climate forecasting as a challenging, high-resolution, problem but their model is far from handling anything related to weather forecasting and their experiments also does not include anything for climate forecasting*
>
> Even though the reviewer makes a good point, it is unfortunately a little outside the scope of the
> paper for the time being. The system of PDEs that would be required to tackle weather forecasting is far more challenging than the benchmark PDEs we are studying here to test our method, and would ask for an entire other study. We first want to evaluate the performance of our methodology on relatively simple settings before we can move to more complex PDEs, which we therefore consider for future work.
>
>
> >*4- Additional Comments*
>
> >*Related Work:*
>
> >*The related work section is written a bit mechanistically and does not highlight some of the distinctive differences of the prior methods in contrast to the authors’ proposed method.*
>
> We re-wrote this part to make sure we highlighted the added value/differences of our methodology with respect to the priori work at the end of each subsection of the related work section.
>
> >*Results:*
>
> >*It would be good if the authors could provide a summary of the results at the beginning of this section.*
>
> We added a short summary of the results in the main text of the paper, at the beginning of the results section, as follows:
>  - **1D case**: Our model MAgNet is able to robustly generalize to unseen meshes in the regular case as compared to the SOTA models FNO and MPNN. The performance is even better in the irregular case
>  - **2D case**: Our model MAgNet is less competitive than MPNN when it comes to generalizing to unseen meshes in the regular case. However, in the irregular case, we are more competitive especially when trained on sparse meshes.
>
> >*Questions:*
>
> >*It is not clear how/when the proposed model fails. Have the authors tested their model's limits?*
>
> There is a limitation section already present in the original manuscript. However, the 2D setting newly studied during the rebuttal made us update the limitations. The final limitations of the method are ultimately the following:
>  - The interpolation seems to only bring little advantage over cubic interpolation when it comes to regular meshes.
>  - It seems that it is less performant in 2D than in 1D, when compared to SOTA methods, particularly for regular meshes (while it is better for irregular meshes)
>  - It shows good super-resolution capabilities, but again significantly better in 1D than in 2D regarding regular meshes.
>  - The forecasting module could be improved to potentially be more stable on long prediction horizons.
>
> >*Limitations:*
>
> >*I am personally hesitant to question societal impacts of any scientific discovery that is not purposefully aiming to promote certain views/directions that directly and objectively harm the society. This is because I believe in the long-term this limits the natural evolution of discovery of the Truth, similar to how protection of the free expression principle leads to democracy. Arguably, any genuine scientific discovery could be eventually used for negative purposes. Therefore, I do not think the reviewers’ subjective opinion should be used in any way as an objective to decide the direction of scientific discoveries.
> Regardless, I do not think the proposed method would have any direct negative impact in the society.*
>
> We appreciate the reviewer’s opinion and consider it as an additional feedback solely to improve the quality of our paper. We do not believe that it will restrain the evolution of the research related to the contents of this paper in any way, yet we again really appreciate the reviewer’s concerns on the matter!

---

> > ### Comment · Area_Chair_wu1E · 2022-08-07
> > **Any feedback?**
> >
> > Dear Reviewer yyq7, authors have provided feedback. Any chance for "live discussion"? I find it really interesting in most of the cases.

---

> > ### Comment · Reviewer_yyq7 · 2022-08-08
> > **Thank you for your response and revising the paper**
> >
> > I appreciate the authors for taking the time to run new experiments and revise the paper in response to the reviews. Please find a my notes about the authors' comments below:
> >
> >     Even though the reviewer makes a good point, it is unfortunately a little outside the scope of the paper for the time being. The system of PDEs that would be required to tackle weather forecasting is far more challenging than the benchmark PDEs we are studying here to test our method, and would ask for an entire other study. We first want to evaluate the performance of our methodology on relatively simple settings before we can move to more complex PDEs, which we therefore consider for future work.
> >
> > I think the authors misunderstood my comment about weather forecasting, and sorry if I was not clear enough. I meant to convey that the writing could look more scientific if the authors had not mentioned climate forecasting in the Abstract, Introduction and Related Works despite the fact that the proposed model cannot address climate forecasting. This might also unintentionally mislead future [young] researchers/readers and take their time, before they realize what the paper is not actually tackling climate forecasting. The authors could talk about such specific long-term goals like solving the climate forecast problem in Discussion or Future Work and talk about the importance of PDE solves in other parts of the paper in a more general way.

---

> > > ### Author Response · Authors · 2022-08-09
> > > **Response to reviewer**
> > >
> > > We thank the reviewer for the feedback and the clarification on the comment regarding weather forecasting - we apologize for the confusion, it is now clear.
> > >
> > > The goal was to use climate forecasting merely as an example of motivation for fast and accurate PDE solving on regular and irregular meshes, the same way we mentioned quantum mechanics or fluid dynamics in the introduction, as important domains of application. As a result, we did mention it in multiple sections as a very practical case (and a very important one) for which our method or another SOTA method could be useful in the future to accelerate numerical solvers, yet never specifically state it as a case study of our paper.
> > >
> > > We do understand the reviewer's concern however, and therefore slightly modified the passages where we mention it (in the Abstract and Related Works) to clarify that is is just a motivation example, and (unfortunately) not an application that we tackle in the paper.

---

### Official Review · Reviewer_4g6A · 2022-07-12

**Rating:** 5
**Confidence:** 4
**Soundness:** 2 fair
**Presentation:** 3 good
**Contribution:** 2 fair

**Summary:**

This paper introduces an architecture for learning mesh-agnostic neural PDE solvers. The architecture consists of encoding, interpolation and forecasting, trained via interpolation loss and forecasting loss. The authors show in a 1D PDE dataset, it outperforms state-of-the-art model MPNN and FNO most of the time in terms of super-resolution on regular and irregular meshes. The authors also investigated how different aspects of the model influences the performance.

**Questions:**

No questions.

**Limitations:**

The main limitation is indicated in the above weaknesses part.

**Strengths And Weaknesses:**

Strengths:

The paper tackles an important problem of learning neural PDE solvers which go beyond the resolution it is trained on. The paper is written clearly. The proposed method makes sense.

Weaknesses:

The most important weakness in this paper is its limited evaluation, which only evaluates in 1D PDEs. Since the paper's main contribution is "Mesh Agnostic", we should see experiments where how the method is evaluated in 2D meshes, which is the standard understanding of a mesh. In 1D, a mesh reduces to connected line segments. Compared with 1D, 2D-mesh introduces many new aspects of difficulty, such as more diversity and variations in the shape of the cells, which leads to more complex interpolations, etc. The results in 1D may not generalize to 2D. I would expect having experiments in 2D mesh, both in a regular grid and a irregular mesh, to see how the model compare with SOTA (e.g. using a 2D experiment where FNO or MPNN/MeshGraphNet is evaluated on but evaluated on super-resolution). Also, in 2D where problems are more difficult, the strength of the method compared to baselines may better exhibit.

--
Update

The authors have addressed my main concerns in the revision, which add experiments in 2D. In 2D experiments, the MPNN still outperforms the proposed method in the Fig. 15 in all cases, which may not show the relative strength of the proposed method. In light of the above, I have updated my rating from 4 to 5.

---

> ### Author Response · Authors · 2022-08-02
> **Official Response to Reviewer 4g6A**
>
> We thank the reviewer for the thoughtful feedback and suggestion regarding evaluation on 2D PDEs. It is indeed a very valid point. We therefore added substantial experiments on 2D PDEs in addition to the original results on the 1D ones, as detailed below.
>
>
> >*The most important weakness in this paper is its limited evaluation, which only evaluates in 1D PDEs. [...] Also, in 2D where problems are more difficult, the strength of the method compared to baselines may better exhibit.*
>
> We added multiple experiments and corresponding results and visualizations on 2D Burger’s equation with 2 choices of parameters, resulting in new datasets that we call B1 and B2. Following the structure of the paper, we also try our method on both regular and irregular meshes, and concerning the irregular mesh, we try 2 settings: one uniform and one non-uniform (normally distributed around a point in the mesh).
>
> The results are as follows:
>
> **B2 PDE, on Regular mesh, trained on a 64x64 resolution:** Figure 15 (b) in the Appendix
>
> **B1 PDE, on Regular mesh, trained on a 64x64 resolution:** Figure 15 (a) in the Appendix
>
> We can observe in that in the regular mesh setting, MPNN surprisingly seems to be the most performant regarding 2D predictions. MAgNet[GNN] also does significantly better than MAgNet[CNN], even though we remain in a regular setting, as it is theoretically able to capture more of the spatial correlations between the nodes. Indeed, leveraging interactions between points after the interpolation module allows MAgNet[GNN] not only to make good predictions but also to generalize to denser and unseen resolutions. Note that while MAgNet[GNN] is not very far from MPNN in terms of performance, it is still quite interesting that the performance shifted when going from 1D to 2D.
> Also surprisingly, FNO performs quite poorly when it comes to generalizing to unseen resolutions.
>
> **B1 PDE on irregular uniform mesh, for different train and test resolutions**: Figure 18 (a) in the Appendix
>
> **B1 PDE on irregular condensed mesh, for different train and test resolutions** : Figure 18 (b) in the Appendix
>
> In the irregular setting however, whether for a sparse or uniform irregular mesh, our MAgNet method appears to retain its superiority, except for few resolutions. Note that the performance of MAgNet[GNN] appears quite robust across different resolutions used in training time, while MPNN strangely appears significantly less performant for some resolutions (specially with a train resolution of 64 and 256).
> We notice that our model MAgNet[GNN] is especially more appealing in case we have fewer points to work with. In the "Condensed" (normally distibuted) setting which is more realistic than the uniform one, MAgNet[GNN] is even more competitive than MPNN. We note however, that while MAgNet[GNN]’s performance is better for small test resolutions, it quickly deteriorates as we test on higher resolutions which is a main limitation of our approach in the 2D case.
>
> Note that 2D visualizations of the predictions at different time steps and for different resolutions are shown in the newly uploaded paper (Appendix A.1). Also, we will add videos showing the 2D rollout predictions as supplementary material in the future, so that readers have a better and more natural visualization of comparison of the different methods in the 2D setting.

---

> > ### Comment · Reviewer_4g6A · 2022-08-07
> > **Response**
> >
> > Thanks the authors for putting the efforts to add the 2D experiments and I have updated my score. From the results in 2D, what do you think is the reason that MPNN outperforms your method in all the resolution (Fig. 15)? For example, do you train both models with the same learning schedule? Can you also provide the total number of parameters for each model (as the number of parameters can influence the expressivity and performance)?
> >
> > For Fig. 18, the MPNN's performance varies a lot, and ideally the performance of resolution 256 should be between 128 and 512. This can be due to variations of training or seed. Maybe the authors can for some experiment scenarios, run the same setting but with different seeds, to verify that MPNN do have a large variation?

---

> > > ### Author Response · Authors · 2022-08-09
> > > **Response to reviewer**
> > >
> > > Many thanks to the reviewer for the response, and for acknowledging our efforts and updating the score as a result.
> > >
> > > We were also quite surprised that MPNN was doing so well in 2D in all resolutions regarding regular meshes, and the reason for such a good performance in 2D as opposed to 2D is not very clear to us. Although both methods leverage message-passing abilities, it seems that our interpolation module really shows advantages when handling irregular meshes.
> > >
> > > >*For example, do you train both models with the same learning schedule?*
> > >
> > > Yes we have! So the difference in performance should not have to do with the scheduler.
> > >
> > > >*Can you also provide the total number of parameters for each model (as the number of parameters can influence the expressivity and performance)?*
> > >
> > > MPNN is trained with 520K parameters, while ours is trained with 2.6M parameters. There is therefore a significant difference of size between the two, as a result of the multiple encoding layers we employ in our model (which show their use in an irregular setting).
> > >
> > > >*For Fig. 18, the MPNN's performance varies a lot, and ideally the performance of resolution 256 should be between 128 and 512. This can be due to variations of training or seed. Maybe the authors can for some experiment scenarios, run the same setting but with different seeds, to verify that MPNN do have a large variation?*
> > >
> > > We were also surprised by the high volatility of the performance of MPNN in this irregular setting.
> > > We have used the same seeds for all the experiments; we therefore do not believe this is not the source of such volatility.
> > >
> > > Yet, in order to test the sensitivity of the model with respect to the seed, we have re-run the experiment of Fig 18 for MPNN with a different seed. Here are the results of this experiment:
> > >
> > > Condensed (normally distributed)
> > > |     | 64x64          | 128x128          | 256x256        | 512x512          |
> > > |-----|----------------|------------------|----------------|------------------|
> > > |N=64 | 0.0617 +- 0.008| 0.069 +- 0.01    | 0.074 +- 0.011 | 0.077 +- 0.012   |
> > > |N=128| 0.0181 +- 0.001| 0.0248 +- 0.001  | 0.0305 +- 0.001| 0.0342 +- 0.001  |
> > > |N=256| 0.1692 +- 0.024| 0.1782 +- 0.027  | 0.1840 +- 0.028| 0.1878 +- 0.029  |
> > > |N=512| 0.0417 +- 0.005| 0.0501 +- 0.006  | 0.0561 +- 0.006| 0.0597 +- 0.007  |
> > >
> > > Uniform:
> > > |     | 64x64          | 128x128          | 256x256        | 512x512          |
> > > |-----|----------------|------------------|----------------|------------------|
> > > |N=64 | 0.207 +- 0.0272| 0.214 +- 0.0307  | 0.212 +- 0.0334| 0.223 +- 0.0348  |
> > > |N=128| 0.0345 +- 0.004| 0.0422 +- 0.005  | 0.0485 +- 0.005| 0.0524 +- 0.006  |
> > > |N=256| 0.0333 +- 0.001| 0.0369 +- 0.0015 | 0.0449 +- 0.002| 0.0485 +- 0.002  |
> > > |N=512| 0.0255 +- 0.002| 0.0316 +- 0.002  | 0.0369 +- 0.002| 0.0524 +- 0.002  |
> > >
> > > When compared with results of Fig 18, it can be observed that the performance of MPNN is here very similar as it is in the paper with original seed, for all resolutions. In particular, the lowest performance observed for the train resolutions of 64x64 and 256x256 for respectively Uniform and Condensed settings remains. It therefore seems rather robust with respect to the seed, and there should be a deeper reason for this variability, that may have to do with the chosen data we use for the experiments.

---

> > > > ### Comment · Reviewer_4g6A · 2022-08-09
> > > > **Response**
> > > >
> > > > Thanks the authors for the reply and additional experiments. Can the authors summarize the benefit/advantage of the proposed method, compared to MPNN, in light of the experiment result of 1D and 2D? To scale the method to even larger application scenarios (e.g. 3D, climate, etc.), it is possible and what main obstacles may lie ahead?

---

> > > > > ### Author Response · Authors · 2022-08-09
> > > > > **Response to reviewer**
> > > > >
> > > > >
> > > > >
> > > > >
> > > > > We thank again the reviewer for the additional feedback.
> > > > >
> > > > > The main advantage of MagNet is that it is generally **more performant in irregular meshes**, as it can be observed for both 1D and 2D experiments (specially with regards to MPNN, as it is more performant than FNO on almost all counts, except for PDE E3 in a 1D regular setting). We added a note in the conclusion to highlight this more.
> > > > >
> > > > > More specifically:
> > > > > - In **1D, it is more performant for almost all training and testing resolutions**, and therefore shows better super/resolution capabilities
> > > > > - In **2D**, MagNet is indeed not showing the best performance in all cases, but it shows **better robustness across the different super/resolution tasks**, which is not the case for MPNN, and is often rather comparable to MPNN even in worst cases. Also, it is interesting to note that **MagNet seems to do significantly better when handed the smallest amount of data to work with** (with a 64x64 training resolution), even in the Condensed non-uniform case. This is a very desirable property for real-life data.
> > > > >
> > > > > Note that the architecture of the model was designed originally with this goal in mind, notably the interpolation module embedded within a learned feature space.
> > > > >
> > > > > In regular meshes, however, it does show a slightly lower performance in 2D regular setting, while still maintaining the best performance in a 1D setting. It is therefore an aspect of the method that we wish to improve in the future, to make it more scalable for traditional regular meshes. A potential future direction lies in the modification of the interpolation module with a more complex strategy than a weighted Nearest-Neighbor-based MLP, in order to capture the spatial interactions more robustly in higher dimensional settings. It might however increase the computational need and therefore the training and inference times, which we will have to keep in mind.

---

### Official Review · Reviewer_XU9U · 2022-07-13

**Rating:** 6
**Confidence:** 3
**Soundness:** 3 good
**Presentation:** 3 good
**Contribution:** 2 fair

**Summary:**

This paper proposes an algorithm to solve PDEs that's agnostic to the input mesh. Precisely, the proposed method follows the "encoder-interpolation-forcast" pipeline. This pipeline will first extract features for each of the sampled mesh points, then these features will be passed into the interpolator to create features for the query points. The query points features will passed through the forcaster to make the final prediction. The experimental results show that MAgNet is capable of performing un par with baselines such as FNO and MPNN. The results also show the proposed interpolation in feature space does lead to improvement. The results also show that the model is able to test on a mesh resolution that's different from the trained mesh resolution.

**Questions:**

In figure 4(a), is there any intuition why the performance is best at resolution 50? How significant is the the gap between cubic and the proposed method?

Is there any visualization of the input and testing mesh? I think it would be helpful to see some of the visualization on some problem setting to provide better intuition for readers.

**Limitations:**

The limitation is properly addressed in section 5.

**Strengths And Weaknesses:**

Strengths:
I think the main strength of the proposed idea is that they did interpolation and interaction in a learned feature space instead of in the physics space directory. This allows the encoder and the interpolator to work together to make a potentially better interpolation schema which is agnostic to the sampling of the mesh vertices. The ablation studies seem to suggest that the learned interpolation is indeed bringing some advantages (per figure 4(a)). I suspect such advantage will be larger (i.e. comparing to Cubic interpolation on the original frames) when it comes to more irregular input meshes.

Weakness:
1. Is the model actually "mesh agnostic"? My understanding of "mesh-agnostic" is that if both meshes represent the same geometry, then the method should output the same results. The experiments seem to show that if two meshes are representing the same geometry, then the method will output equally accurate results, which is a weaker guarantee. It's not clear to me that the encoder architecture will provide exactly the same feature if the input mesh is slightly changed.

2. How well is the method robust to irregular mesh? One major concern I have toward this method is that, if the input mesh is sample in a extremely non-uniform way (which seems to be the case for many physics problem as higher numerical precision will be required in one place than the other), this will require the interpolation to work with different descretization density, which might potentially lead to interpolation error. The experiment regarding irregular mesh is in section 4.2. There is very little description about how the irregular mesh is sampled (or visualization). Also to the bet of my understanding, it's only tested on the irregular mesh, but not using irregular mesh as input. I suspect this also seems to be related to the limitation of the ablation study on interpolation schema, where the difference between the cubic interpolation is not much larer than that of the feature space interpolation. More clarification toward this point will be helpful.

---

> ### Author Response · Authors · 2022-08-02
> **Official Response to Reviewer XU9U**
>
> We thank the reviewer for the thoughtful feedback and suggestions to improve the paper. It resulted in a significant amount of updates within the original draft, which can be observed in the updated uploaded PDF of the paper.
>
> We respond to the reviewer’s questions/comments below, indicating a change in the paper when it applies.
>
> >*Strengths: I think the main strength of the proposed idea is that they did interpolation and interaction in a learned feature space instead of in the physics space directory. [...] I suspect such advantage will be larger (i.e. comparing to Cubic interpolation on the original frames) when it comes to more irregular input meshes.*
>
> Indeed, we demonstrate in new experiments that our method shows particularly good performance with respect to compared other methodologies on irregular meshes, as discussed further in the second “Weakness” comment below. Thanks again for the suggestion.
>
> >*Weakness:*
>
> >*1. Is the model actually "mesh agnostic"? [...] It's not clear to me that the encoder architecture will provide exactly the same feature if the input mesh is slightly changed.*
>
> The reviewer makes a good point that we wish to discuss and clarify. If the model was supposed to represent the solution of the PDE, then indeed, the model should output the same value for a point belonging to any arbitrary mesh. However, our model acts as a **solver** rather than an **exact PDE solution estimator**. This means that naturally, as the mesh changes (e.g. becomes denser around some point of interest), we expect the solution at that point to have a different value than if it was evaluated on a coarser mesh. Think of how a numerical PDE solver would output different values for a certain point in the PDE domain if the mesh changes: that’s exactly what we replicate here.
>
> Our first attempt was to do exactly as the reviewer suggested: produce solutions that would have the same value at some point regardless of the mesh. However, it prevented us from gaining super-resolution capabilities (see Table (2) in the manuscript ). Introducing dependencies between points in the test mesh allowed us to be more robust to changes in the mesh but indirectly forced our model to act more like a solver than a PDE solution estimator.
>
> >*2.How well is the method robust to irregular mesh? [...] More clarification toward this point will be helpful.*
>
> That is again a very good point. We included additional experiments with non-uniform (but normally distributed) irregular meshes for the 2D experiments we added, therefore testing the ability to interpolate within meshes characterized by a varying density of points.
>
> The results regarding the B1 PDE data, also included in the updated PDF of the article in Figure 18 in the Appendix.
>
> The performance of MAgNet[GNN] appears quite robust across different resolutions used in training time, while MPNN strangely appears less performant for some resolutions (specially with a train resolution of 64 and 256). It seems generally more performant that MPNN, except for a few choices of test resolutions. This setting therefore shows the relative superiority of MAgNet in a sparse irregular configuration (which indeed often happens with real-life data)
>
> Regarding your comment on using irregular meshes as input: MAgNet[GNN] uses a GNN as an encoder, so it’s effectively able to handle irregular meshes as input as well. Therefore, when using MAgNet[GNN] with irregular meshes, we indeed use the irregular mesh as the input.
>
> >*Questions:*
>
> >*In figure 4(a), is there any intuition why the performance is best at resolution 50? How significant is the the gap between cubic and the proposed method?*
>
> Given that the training resolution was 50, it is natural that it is an easier task for the model to achieve a good prediction on the same resolution rather than doing it while performing required under/super-resolution. It is therefore not surprising that this resolution offers the best performance.
> It can be seen from the sensitivity study on the interpolation method, as can be observed from Figure 3a, that the performance is rather comparable between the cubic interpolation, and our interpolation module.
>
> >*Is there any visualization of the input and testing mesh? I think it would be helpful to see some of the visualization on some problem setting to provide better intuition for readers.*
>
> Thank you for the nice suggestion. We added a visualization of the regular and irregular meshes, for the 2D PDEs (which we added during the rebuttal), as we believe 2D plots are often more natural to readers. The visualization can be seen in Figure **7** and **8** in the Appendix.

---

> > ### Comment · Area_Chair_wu1E · 2022-08-07
> > **Any feedback?**
> >
> > Dear Reviewer XU9U, authors have provided feedback. Any chance for "live discussion"? I find it really interesting in most of the cases.

---

### Author Response · Authors · 2022-08-02
**Common answer to all reviewers**

We thank the reviewers for their efforts and reviews of high quality. We are happy that they appreciated our contributions to the PDE solving literature, particularly focusing on the ability to work on irregular meshes and perform zero-shot super resolution.

We carefully considered your suggestions, and believed that it ultimately made our paper significantly better; we therefore thank you again for it. We answered each reviewer separately, point by point, using the discussion tool. In a nutshell, these are the main points and  additions made to the original manuscript:
- We addressed the concerns with the “mesh-agnostic” denomination.
- We added substantial experiments with a totally newly generated dataset for 2D PDEs (2D Burgers equation) to test the performance of our method in a 2D setting, and see how it compares to SOTA methods in this setting (often not considered in many publications). (**Appendix A.4; new Figures 15 and 18**)
- Following the previous points, we added visualizations of 2D predictions for multiple time steps, for multiple zero-shot super resolution settings, for both regular (**Appendix A.1; new Figures 7, 8, 9 and 10**) and irregular meshes. (**Appendix A.1; new Figures 11, 12, 13, and 14**)
- We added tests with highly non-uniform irregular meshes (normally distributed) to test the performance of our method in varying point density settings (in 2D). (**Appendix A.4; new Figure 18**)
- We added visualizations of irregular input and test meshes (uniform and non-uniform) so that the reader can have a better intuition of what it represents. (**Appendix A.3; new Figures 16 and 17**)
- Added the lower performance of our MAgNet in 2D regular meshes as a limitation in the limitations section of the article.

We found that while our MAgNet method is more performant than SOTA methods in 1D, especially for irregular meshes (and when performing zero-shot super resolution), its superiority is indeed not so clear in the 2D setting. While it appears marginally less competitive than MPNN when it comes to generalizing to unseen meshes in the 2D regular case, MAgNet is more competitive in the irregular case, especially when trained on sparse (highly non uniform) meshes.

While discussing new results here in the discussion, we will refer to the corresponding figure with the updated number in the new uploaded PDF. In addition, more details and comments on the new results can be found in the newly updated manuscript.

---

### Meta-Review · Area_Chair_wu1E · 2022-08-26

**Recommendation:** Accept
**Confidence:** Certain

**Metareview:**

The paper proposes an architecture that maps from an input function to an output function that can handle unstructured meshes. On a set of extensive experiments the effectiveness and robustness of the model is shown. As in other models (like Fourier Neural Operator) the PDE itself is not present in the loss, so 'solution' has to be used with caution. However, the overall design and presentation is impressive, and there is a general agreement between the reviewers about the importance of the work.




**Award:**

No

---

### Decision · Program_Chairs · 2022-09-14

Accept